# Gradient Descent: The Ultimate Optimizer

**Kartik Chandra**[*]
MIT CSAIL[†]
Cambridge, MA
kach@csail.mit.edu

**Audrey Xie**[*]
MIT CSAIL
Cambridge, MA
ahx@mit.edu

**Jonathan Ragan-Kelley**
MIT CSAIL
Cambridge, MA
jrk@csail.mit.edu

**Erik Meijer**
Meta, Inc.
Menlo Park, CA
erikm@fb.com

## Abstract

Working with any gradient-based machine learning algorithm involves the tedious task of tuning the optimizer's hyperparameters, such as its step size. Recent work has shown how the step size can itself be optimized alongside the model parameters by manually deriving expressions for "hypergradients" ahead of time.

We show how to *automatically* compute hypergradients with a simple and elegant modification to backpropagation. This allows us to easily apply the method to other optimizers and hyperparameters (e.g. momentum coefficients). We can even recursively apply the method to its own *hyper*-hyperparameters, and so on *ad infinitum*. As these towers of optimizers grow taller, they become less sensitive to the initial choice of hyperparameters. We present experiments validating this for MLPs, CNNs, and RNNs. Finally, we provide a simple PyTorch implementation of this algorithm (see people.csail.mit.edu/kach/gradient-descent-the-ultimate-optimizer).

## 1   Introduction

When we train deep neural networks by gradient descent, we have to select a step size $\alpha$ for our optimizer. If $\alpha$ is too small, the optimizer runs very slowly, whereas if $\alpha$ is too large, the optimizer fails to converge. Choosing an appropriate $\alpha$ is thus itself an optimization task that machine learning practitioners face every day. Why not apply gradient descent here, too? To do so, we need to compute the derivative of the loss function not only with respect to the neural network's weights, but also with respect to $\alpha$. Baydin et al. (2018), applying an insight from Almeida et al. (1999), describe how to efficiently compute such "hypergradients" by manually differentiating standard optimizer update rules with respect to the step size hyperparameter. This allows for on-line learning rate adaptation, which generally improves convergence, especially when the initial $\alpha$ is sub-optimal.

However, the above method has three limitations: (1) manually differentiating optimizer update rules is tedious and error-prone, and must be re-done for each optimizer variant; (2) the method only tunes the step size hyperparameter, not other hyperparameters such as the momentum coefficient; and (3) the method introduces a *new* hyperparameter, the hyper-step-size, which must also be tuned.

In this paper, we address all three limitations by replacing manual differentiation with *automatic differentiation* (AD), which (1) automatically computes correct derivatives without any additional human effort, and (2) naturally generalizes to other hyperparameters (e.g. momentum coefficient) for free. As for (3), we observe that AD can be applied to optimize not only the hyperparameters, but also the *hyper*-hyperparameters, and the hyper-hyper-hyperparameters, and so on. In fact, we can implement arbitrarily tall towers of recursive optimizers, which are increasingly robust to the choice of initial hyperparameter. These "hyperoptimizers" therefore reduce the burden on humans responsible for tuning the hyperparameters. (Such an effect was hypothesized by Baydin et al., but not tested because manual differentiation of complex sequences of nested optimizers is impractical.)

---

[*]Equal contribution.
[†]Work done in part at Meta, Inc. and in part at Stanford University.

36th Conference on Neural Information Processing Systems (NeurIPS 2022).

Although "just apply AD" is a seemingly straightforward recipe, an efficient implementation that properly allows for recursive self-application requires some care. To close the loop, we take inspiration from the study of recursion and combinators in programming language theory (and the long tradition of programming language papers named "Lambda: The Ultimate X"). We spell out the details in Section 2, and evaluate our method in Section 3. We find that across a variety of architectures (MLPs, CNNs, and RNNs) our hyperoptimizers are robust to suboptimal choices of initial hyperparameters, and that this robustness increases as we grow the stacks of optimizers taller.

## 2 Implementing hyperoptimizers

Consider some loss function $f$ that we want to minimize using gradient descent, and let $w_i$ be the weights at the beginning of step $i$ (we will omit subscripts on $f$, even though it varies at each step due to the stochasticity of minibatches). First, recall the standard weight update rule at step $i$ for SGD, using some fixed step size $\alpha$:

$$w_{i+1} = w_i - \alpha \frac{\partial f(w_i)}{\partial w_i}$$

We would like to also update $\alpha$ at each step, so we will index it as well with the step number; that is, let $\alpha_i$ be the step size at the beginning of step $i$. At each step, we will first update $\alpha_i$ to $\alpha_{i+1}$ using some update rule yet to be derived, and then use the updated step size $\alpha_{i+1}$ to update the weights from $w_i$ to $w_{i+1}$.

$$\alpha_{i+1} = \alpha_i - \boxed{\text{adjustment for } \alpha_i}$$
$$w_{i+1} = w_i - \alpha_{i+1} \frac{\partial f(w_i)}{\partial w_i}$$

What should the adjustment for $\alpha_i$ be? By analogy to $w$, we want to adjust $\alpha_i$ in the direction of the gradient of the loss function with respect to $\alpha_i$, scaled by some *hyper*-step size $\kappa$. In other words, the adjustment should be $\kappa(\partial f(w_i)/\partial \alpha_i)$. Our modified update rule is therefore:

$$\alpha_{i+1} = \alpha_i - \kappa \frac{\partial f(w_i)}{\partial \alpha_i} \tag{1}$$

$$w_{i+1} = w_i - \alpha_{i+1} \frac{\partial f(w_i)}{\partial w_i} \tag{2}$$

All that remains is to compute $\partial f(w_i)/\partial \alpha_i$ in equation (1). Below, we first review how Baydin et al. (2018) take this derivative by hand. Then, we show how to obtain the same result automatically and efficiently using AD. Finally, we discuss how this automation allows us to generalize the method.

### 2.1 Computing the step-size update rule by hand

One option to compute $\partial f(w_i)/\partial \alpha_i$, explored by Baydin et al. (2018), is to proceed by direct manual computation of the partial derivative. Applying the chain rule to (1), we have

$$\frac{\partial f(w_i)}{\partial \alpha_i} = \frac{\partial f(w_i)}{\partial w_i} \cdot \frac{\partial w_i}{\partial \alpha_i} = \frac{\partial f(w_i)}{\partial w_i} \cdot \frac{\partial \left( w_{i-1} - \alpha_i \frac{\partial f(w_{i-1})}{\partial w_{i-1}} \right)}{\partial \alpha_i} \tag{3}$$

$$= \frac{\partial f(w_i)}{\partial w_i} \cdot \left( -\frac{\partial f(w_{i-1})}{\partial w_{i-1}} \right) \tag{4}$$

where (3) is obtained by substituting the update rule in (2) for $w_i$ and (4) is obtained by observing that $w_{i-1}$ and $f(w_{i-1})$ do not depend on $\alpha_i$. As Baydin et al. note, this expression lends itself to a simple and efficient implementation: simply remember the past two gradients from backpropagation, and take their dot product to obtain the hypergradient with respect to the step size.

We were able to take this derivative by hand because the update rule for SGD is simply a multiplication by a constant, whose derivative is trivial. What about other optimizers? Consider the Adam optimizer (Kingma and Ba, 2014), which has a much more sophisticated update rule involving the four hyperparameters $\alpha, \beta_1, \beta_2, \epsilon$ (though $\epsilon$ is typically not tuned). Differentiating the update rule by hand,

we obtain *significantly* more complex expressions for the hypergradients:

$$\frac{\partial w_i}{\partial \alpha_i} = -\frac{\hat{m}_i}{\left(\epsilon_i + \sqrt{\hat{v}_i}\right)} \qquad \frac{\partial w_i}{\partial \beta_{1i}} = -\frac{\alpha_i \left(-\frac{\partial f(w_{i-1})}{\partial w_{i-1}} + m_{i-1} + i\beta_{1i}^{(i-1)}\hat{m}_i\right)}{\left(1 - \beta_{1i}^i\right)\left(\epsilon_i + \sqrt{\hat{v}_i}\right)}$$

$$\frac{\partial w_i}{\partial \epsilon_i} = \frac{\alpha_i \hat{m}_i}{\left(\epsilon_i + \sqrt{\hat{v}_i}\right)^2} \qquad \frac{\partial w_i}{\partial \beta_{2i}} = \frac{\alpha_i \hat{m}_i \sqrt{\hat{v}_i}\left(-\left(\frac{\partial f(w_{i-1})}{\partial w_{i-1}}\right)^2 + v_{i-1} + i\beta_{2i}^{(i-1)}\hat{v}_i\right)}{2v_i\left(\epsilon_i + \sqrt{\hat{v}_i}\right)^2}$$

This manual approach to derive hypergradients simply does not scale: it is tedious and error-prone, and must be repeated for every optimizer variant. However, with a little bit of care, we can compute hypergradients automatically and efficiently alongside the regular gradients.

## 2.2 Computing the step-size update rule automatically

In order to compute hypergradients automatically, let us first briefly review the mechanics of reverse-mode AD. Differentiable programming systems that provide reverse-mode AD typically build up a computation graph as the function is computed forwardly. For example, when a user computes the function $f(w_i)$, the system internally stores a DAG whose leaves are the weights $w_i$, whose internal nodes are intermediate computations, and whose root is the final loss. It can then backpropagate through the computation graph starting at this root node, depositing gradients in each internal node as it descends, until the weights $w_i$ at the leaf nodes have accumulated their gradients $\partial f(w_i)/\partial w_i$. Once the gradient $\partial f(w_i)/\partial w_i$ is computed by the backwards pass, we update the weights $w_{i+1} = w_i - \alpha \cdot \partial f(w_i)/\partial w_i$, and repeat the cycle for the next step of gradient descent.

An important consideration at this point is for the weights to be "detached" from the computation graph before the next iteration of this algorithm — that is, for the weights to be forcibly converted to leaves of the graph by removing any inbound edges. The effect of the "detach" operation is depicted in Figure 1a. If this step were skipped, backpropagation at the next iteration would continue beyond the current weights and into the previous iteration's computation graph. Over time the computation graph would grow taller linearly in the number of steps taken; because backpropagation is linear in the size of the graph, the overall training would become quadratic-time and intractable.

Let us take PyTorch as an example. In the built-in SGD optimizer (Paszke et al., 2017, optim/sgd.py, commit ff94c9d), this is implemented by wrapping the update in the `@torch.no_grad()` context manager. Here, we need finer grained control over gradient flow, so will make the `.detach()` operations explicit. Below is pseudocode for an SGD optimizer that uses `.detach()` as we have discussed. The highlighted calls to `.detach()` correspond to detaching the weights and their gradients.

```
def SGD.__init__(self, alpha):
  self.alpha = alpha

def SGD.step(w):
  d_w = w.grad.detach()
  w = w.detach() - self.alpha.detach() * d_w
```

Now, in order to have backpropagation deposit the gradient with respect to $\alpha_i$ as well as $w_i$, we can simply refrain from detaching $\alpha_i$ from the graph, detaching instead *its* parents. This is depicted in Figure 1b. Because we want to compute $\partial f(w_i)/\partial \alpha_i$, the edge from $\alpha_i$ to $w_i$ needs to remain intact. To implement this, instead of calling `.detach()` on `alpha` directly, we detach its parents when applying equation (1). This yields the following fully-automated hyperoptimization algorithm:

```
def HyperSGD.step(w):
  # update alpha using equation (1)
  d_alpha = self.alpha.grad.detach()
  self.alpha = self.alpha.detach() - kappa.detach() * d_alpha

  # update w using equation (2)
  d_w = w.grad.detach()
  w = w.detach() - self.alpha.detach() * d_w
```

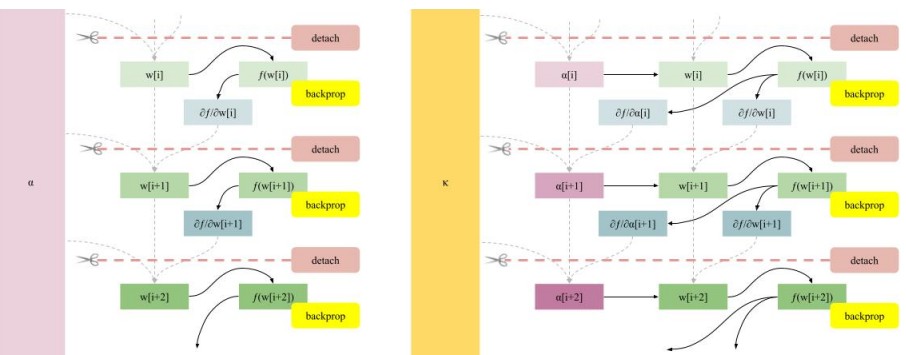

(a) Computation graph of SGD with a single fixed hyperparameter $\alpha$.

(b) Computation graph of SGD with a continuously-updated hyperparameter $\alpha_i$.

Figure 1: Visualizing the computation graphs of SGD and HyperSGD.

Since we only extend the computation graph by a little extra amount, corresponding to evaluating the optimizer, the hyperoptimizer's computational overhead is negligible (see Figure 4f).

## 2.3 Extending to other optimizers

As suggested by Maclaurin et al. (2015), it should be possible to apply gradient-based methods to tune hyperparameters of common variations on SGD such as AdaGrad (Duchi et al., 2011), AdaDelta (Zeiler, 2012), or Adam (Kingma and Ba, 2014). The above implementation of HyperSGD generalizes quite easily to these optimizers — we simply replace the last line with the new update rule.

Unlike previous work, our method also allows for simultaneously optimizing *all* hyperparameters of these optimizers (e.g. all of $\alpha$, $\beta_1$, and $\beta_2$ for Adam) "for free." We simply treat them just like `alpha` in the implementation. Our evaluation in Section 3.2 demonstrates that this indeed advantageous to do. There are, however, two important subtleties: First, because hyperparameters like $\beta_1$ and $\beta_2$ must be strictly in the domain $(0, 1)$, we clamp the "raw" values to this domain using a scaled sigmoid. Without this step, we might accidentally adjust these values outside their domains. Second, the Adam optimizer in particular involves the term $\sqrt{\hat{v}_i}$, which is continuous but not differentiable at $\hat{v}_i = 0$. Because Adam normally initializes $\hat{v}_0 = 0$, backpropagation fails on the first step due to division by zero. We fix this problem by initializing $\hat{v}_0$ to $\epsilon$ rather than 0.

## 2.4 Stacking hyperoptimizers recursively

At this point it is natural to ask whether the hyperoptimizer can itself be optimized; that is, whether the hyper-hyperparameters can be adjusted by a hyper-hyperoptimizer. The possibility of doing so recursively *ad infinitum* to obtain an optimization algorithm that is highly robust to the human-chosen hyperparameter was hypothesized by Baydin et al. (2018). Computing the gradients of these higher-order hyperparameters by hand is impossible without knowing the exact sequence of stacked optimizers in advance, and, as discussed above, would be extremely tedious and error-prone.

However, the ability to compute these gradients automatically by AD makes it possible to realize this vision. To do so, let us revisit our previous implementation of HyperSGD. Notice that there is an opportunity for recursion lurking here: the adjustment to `alpha` can be factored out with a call to `SGD.step`, where SGD's hyperparameter is `kappa`.

```python
def HyperSGD.step(w):
  # update alpha using Equation (1)
  SGD(kappa).step(self.alpha)

  # update w using Equation (2)
  d_w = w.grad.detach()
  w = w.detach() - self.alpha * d_w
```

Because SGD is already careful to properly detach its parameter (typically $w$, but in this case $\alpha$), this implementation is functionally identical to the one above. Indeed, any optimizer that observes this protocol would suffice, so let us abstract out the optimizer as a parameter to HyperSGD:

```
def HyperSGD.__init__(self, alpha, opt):
  self.alpha = alpha
  self.optimizer = opt

def HyperSGD.step(w):
  self.optimizer.step(self.alpha)
  d_w = w.grad.detach()
  w = w.detach() - self.alpha * d_w

opt = HyperSGD(0.01, opt=SGD(kappa))
```

Finally, after this refactoring, we can recursively feed `HyperSGD` *itself* as the optimizer, obtaining a level-2 hyperoptimizer `HyperSGD(0.01, HyperSGD(0.01, SGD(0.01)))`. Similarly, we can imagine taller towers, or towers that mix and match multiple different kinds of optimizers, such as Adam-optimized-by-SGD-optimized-by-Adam.

A natural concern is whether this process actually exacerbates the hyperparameter optimization problem by introducing even more hyperparameters. Baydin et al. (2018) predicted that as the towers of hyperoptimizers grew taller, the resulting algorithms would become less sensitive to the human-chosen hyperparameters. This is indeed the case; Section 3.4 presents an empirical evaluation.

## 3  Experiments

In this section we evaluate the hyperoptimizers made possible by our system, exploring in particular the benefits of optimizing hyperparameters beyond just the step size, and of stacking hyperoptimizers to multiple levels. Each of these experiments was conducted on a single NVIDIA TITAN Xp GPU.

### 3.1  Hyperoptimization for SGD

First, we establish some basic properties about hyperoptimizers: (1) whether an SGD hyperoptimizer performs better than a regular SGD optimizer, and (2) whether the final learned step size is better than the initial human-chosen step size. We test the latter property by running a fresh regular SGD optimizer with the final learned step size of the hyperoptimizer. Following authors of prior work (Maclaurin et al., 2015; Baydin et al., 2018), we conducted initial experiments on the MNIST dataset (Lecun et al., 1998) using a neural network with one fully-connected hidden layer of size 128, tanh activations, and a batch size of 256. We trained all networks for 30 epochs, reporting statistics over 3 runs. As a baseline we used SGD with $\alpha = 0.01$.

Table 1a summarizes the results of our experiments. **We find that hyperoptimized SGD outperforms the baseline by a significant margin.** This holds even if we use other optimizers (e.g. Adam) to adjust the step size of the SGD optimizer. Furthermore, when we re-ran the regular optimizers with the new learned hyperparameters, we found that they performed better than the initial hyperparameter.

### 3.2  Hyperoptimization for Adam, AdaGrad and RMSProp

In Section 2.3, we described how to apply our system to the Adam optimizer, simultaneously optimizing not only the learning rate $\alpha$, but also the momentum coefficients $\beta_{1,2}$. Here, we ask three questions: (1) whether hyperoptimized Adam optimizers perform better than regular Adam optimizers, (2) whether the learned hyperparameters outperform the baseline, and (3) whether there is a benefit to optimizing all the hyperparameters, as opposed to only optimizing the learning rate as Baydin et al. (2018) do. Because Adam has significantly faster convergence than SGD, we only run these experiments for 5 epochs to avoid overfitting.

Table 1b summarizes the results of our experiments. We find that indeed the hyperoptimized Adam optimizer outperforms the regular Adam optimizer on its "default" settings. As with SGD, the learned hyperparameters perform better than the initial hyperparameters when re-run with the regular optimizer. Inspecting the learned hyperparameters, we find that the algorithm raises the learning rate

| Optimizer | Test error |
|---|---|
| SGD | 8.99±0.05% |
| SGD / SGD
SGD(0.0769) | 4.81±0.10%
5.44±0.10% |
| SGD / Adam(0.1)
SGD(0.4538) | 4.86±0.06%
2.80±0.09% |
| SGD / AdaGrad
SGD(0.0836) | 4.85±0.21%
5.17±0.03% |
| SGD / RMSprop(0.1)
SGD(0.5920) | 4.52±0.02%
2.52±0.07% |

(a) Experiments with SGD (Section 3.1)

| Optimizer | Test error |
|---|---|
| Adam | 4.67±0.06% |
| Adam / SGD($10^{-5}$)
Adam(0.0040, 0.899, 0.999) | 3.03±0.02%
3.11±0.06% |
| $\text{Adam}^\alpha$ / SGD($10^{-5}$)
$\text{Adam}^\alpha$(0.0021) | 3.12±0.04%
3.47±0.02% |
| Adam / Adam
Adam(0.0038, 0.870, 0.999) | 3.05±0.09%
3.24±0.13% |
| $\text{Adam}^\alpha$ / Adam
$\text{Adam}^\alpha$(0.0036) | 3.04±0.08%
3.08±0.12% |

(b) Experiments with Adam (Section 3.2)

| Optimizer | Test error |
|---|---|
| AdaGrad | 7.40±0.08% |
| AdaGrad / SGD
AdaGrad(0.0080) | 6.90±0.16%
7.75±0.02% |
| AdaGrad / AdaGrad
AdaGrad(0.0151) | 5.03±0.23%
6.67±0.08% |

(c) Experiments with AdaGrad (Section 3.2)

| Optimizer | Test error |
|---|---|
| RMSProp | 4.19±0.47% |
| $\text{RMSProp}^\alpha$ / SGD($10^{-4}$)
RMSProp(0.0030) | 3.55±0.23%
3.93±0.70% |
| $\text{RMSprop}^{\alpha,\gamma}$ / SGD($10^{-4}$)
RMSProp(0.0032, 0.9899) | 3.33±0.07%
3.25±0.09% |
| $\text{RMSProp}^\alpha$ / RMSProp($10^{-4}$)
RMSProp(0.0021) | 3.42±0.45%
3.60±0.04% |
| $\text{RMSProp}^{\alpha,\gamma}$ / RMSProp($10^{-4}$)
RMSProp(0.0020, 0.9962) | 2.96±0.11%
3.65±0.36% |

(d) Experiments with RMSProp (Section 3.2)

Table 1: Hyperoptimization experiments with MNIST. We denote hyperoptimizers by their constituent optimizers separated by slashes (the leftmost item adjusts the model's weights). $\text{Adam}^\alpha$ is an Adam optimizer where only $\alpha$ is optimized as by Baydin et al. (2018); $\text{RMSProp}^\alpha$ is similar. If not specified, initial hyperparameters are PyTorch defaults ($10^{-2}$ for learning rates except $10^{-3}$ for Adam; $\beta_1 = 0.9, \beta_2 = 0.99$ for Adam and $\gamma = 0.99$ for RMSProp). Each hyperoptimizer experiment is repeated using the final hyperparameters (typeset in pink) learned by the algorithm.

$\alpha$ and slightly lowers $\beta_1$, but does not significantly affect $\beta_2$. Nevertheless, learning $\beta_1$ does help slightly, though not when the top-level optimizer is itself another Adam optimizer.

Similarly, we can add any other optimizer to our system with just a few straightforward lines of code. Here, we show results for AdaGrad (Table 1c) and RMSProp (Table 1d; also run to 5 epochs). **These experiments took less than an hour each to implement from scratch, and show that every hyperoptimizer stack outperforms the non-hyperoptimized baseline.** We remark that AdaGrad is known to "stall" over time as the effective step size goes to zero; inspecting the learned $\alpha$ over time, we find that the AdaGrad/AdaGrad hyperoptimizer increases $\alpha$ to make up for this effect. Additionally, we tried to hyperoptimize RMSProp's new $\gamma$ parameter, which modulates the accumulation of gradient RMS terms. This yielded even better results (compare $^\alpha$ to $^{\alpha,\gamma}$ trials), and required only a 1-line change in our code.

## 3.3 Hyperoptimization at scale

Next, we evaluate our hyperoptimizers on two different real-world neural network architectures.

### 3.3.1 Convolutional neural networks for computer vision

We train a ResNet-20 (He et al., 2016) with and without hyperoptimization on the CIFAR-10 dataset (Krizhevsky, 2012). As a baseline, we replicate the training procedure of He et al. (2016): we

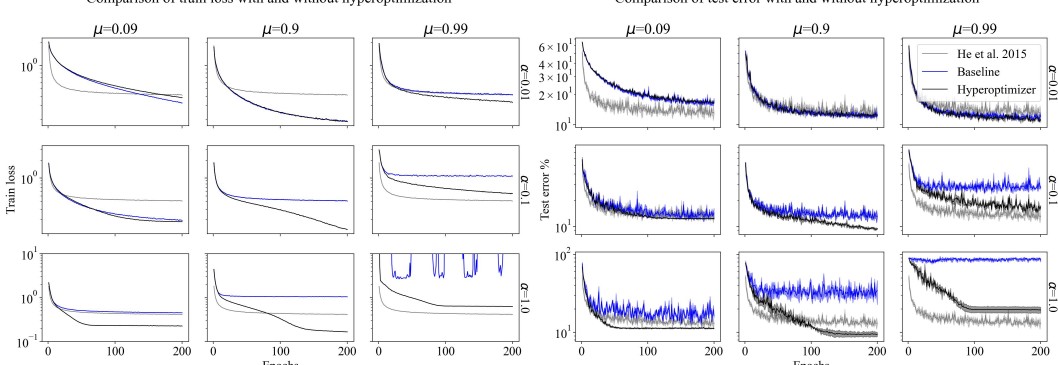

(a) For a wide range of "bad" initial hyperparameter configurations, the hyperoptimizer improves on (or at least matches) final test accuracy, and often matches or even outperforms the "good" initial hyperparameters.

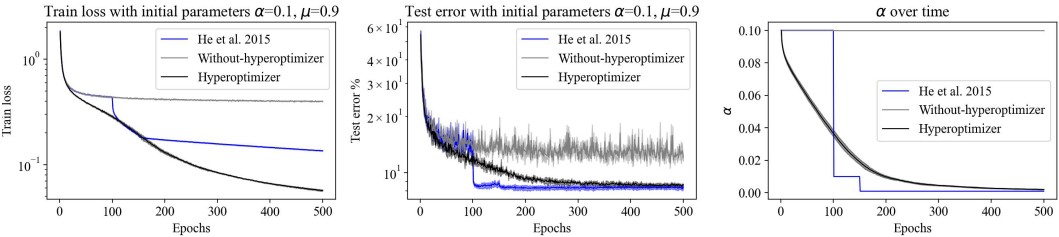

(b) The hyperoptimizer matches performance of the hand-engineered learning rate decay schedule by He et al. (2016), learning a strikingly similar decay schedule (right plot).

Figure 2: Training ResNets on CIFAR-10 with hyperoptimizers (Section 3.3.1).

use the same network architecture, optimizer (SGD), step size (0.1), momentum (0.9), and weight decay ($10^{-4}$), though *without* their custom learning rate decay schedule (which we will address later). Experiments were run for 200 epochs, which takes around 3 hours on our hardware.

First, we test how robust the hyperoptimizer is to "bad" initial choices of step size and momentum. We vary the initial step size and the momentum among "small," "good," and "large" values (that is, $\alpha \in \{0.01, 0.1, 1.0\}$ and $\mu \in \{0.09, 0.9, 0.99\}$), and add a hyperoptimizer ($\alpha_\alpha = \alpha^2 \cdot 10^{-3}$, $\alpha_\mu = 1/(1-\mu) \cdot 10^{-6}$). The results of this experiment are shown in Figure 2a. In every configuration, the hyperoptimizer matches or outperforms the regular optimizer in final test accuracy. Furthermore, in nearly all of the configurations, the hyperoptimizer matches or exceeds the "good" hyperparameters' final test accuracy. Only when *both* hyperparameters are bad in the same direction (too small or too large) is it unable to manage this, and even then for the too-large case it dramatically lowers the loss compared to no hyperoptimizer. **We conclude that hyperoptimizers are indeed beneficial for tuning both step size and momentum in this real-world setting.**

Next, we add in the learning rate decay schedule hand-engineered by He et al. (2016): the step size is divided by 10 at epochs 100 and 150. We compare this with a hyperoptimizer initialized with the same starting hyperparameters, training both variants for 500 epochs. Our results are shown in Figure 2b. **The hyperoptimizer not only matches the final test loss of the hand-engineered learning rate decay schedule, but also learns a decay schedule strikingly similar to one hand-engineered by He et al.** Of course, both networks significantly outperform the baseline trained with a fixed step size.

### 3.3.2   Recurrent neural networks for language modeling

We train a character-level RNN ("Char-RNN") on the Tolstoy dataset, as proposed by Karpathy et al. (2015) as a convenient testbed for language models, which is now often used to benchmark optimizers (Schneider et al., 2018; Schmidt et al., 2021). We took the architecture (2-layer LSTM with 128 hidden nodes) and "expert" optimizer (Adam optimizer with $\alpha = 2 \times 10^{-3}$, run for 50,000 gradient descent steps) directly from Johnson (2017) as recommended by Karpathy et al. We compare against

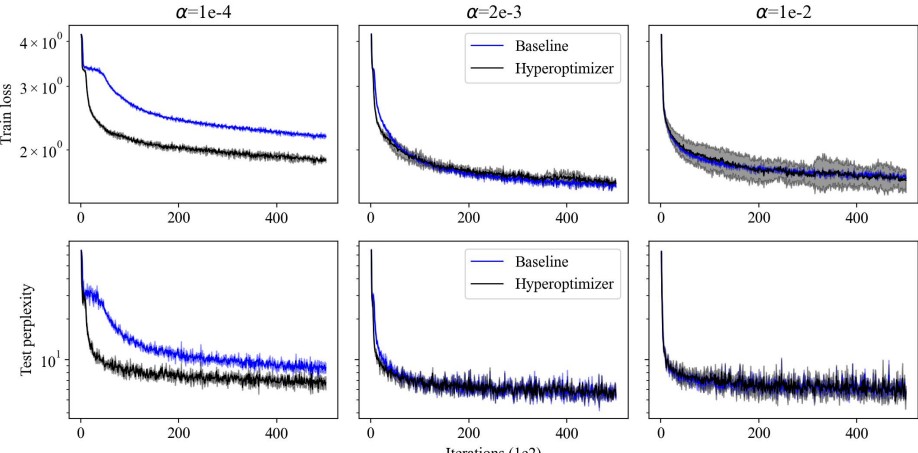

Figure 3: Training RNNs with hyperoptimizers (Section 3.3.2). As the initial learning rate is lowered, the regular Adam optimizer's convergence slows, but the hyperoptimizer is able to accelerate it. The hyperoptimizer also slightly improves convergence when the initial learning rate is too high.

our HyperAdam optimizer on a wide range of initial learning rates $\alpha \in \{10^{-4}, 2 \times 10^{-3}, 10^{-2}\}$, with $\alpha_\alpha = \alpha \cdot 10^{-2}$. We do not vary *initial* $\beta_{1,2}$ because in our experience these hyperparameters are typically left at their default values. However, we *do* allow the hyperoptimizer to vary $\beta_{1,2}$ over the course of training (with $\alpha_{\beta_1} = 10^{-4}$ and $\alpha_{\beta_2} = 2 \times 10^{-4}$). All runs took around 1 hour to train.

The results of this experiment are shown in Figure 3. We find that the hyperoptimizer performs comparably to the expert-chosen fixed step size (perplexity $5.41 \pm 0.26$ with hyperoptimizer vs $5.27 \pm 0.31$ without), and improves upon "bad" initial step sizes in both directions ($5.45 \pm 0.76$ vs $5.77 \pm 0.34$ when too high; $6.51 \pm 0.88$ vs $8.71 \pm 0.91$ when too low).

### 3.4 Higher-order hyperoptimization

In Section 2.4 we developed an interface for building arbitrarily tall towers of optimizers. Baydin et al. (2018) hypothesized that taller towers would yield hyperoptimizers that were increasingly robust to the initial human-chosen hyperparameters. To validate this behavior of higher-order hyperoptimizers, we ran each of our benchmarks from above (MLP on MNIST, CNN on CIFAR-10, RNN on Tolstoy) with towers of hyperoptimizers of increasing heights, and with bottom-level step sizes $\alpha$ initialized across many orders of magnitude. In practice we find that if the initial hyper-step sizes are too large, the computation diverges for networks larger than the MNIST MLP. So, we initialize each level's hyperparameter to be smaller than that of the previous level. Specifically, we use the following scheme: from $\alpha = 10^{-8}$ to $10^{-4}$ the higher layers' step sizes were initialized to $[\alpha \cdot 10^2, \alpha \cdot 10^0, \alpha \cdot 10^{-2}]$ respectively, while for $\alpha \geq 10^{-3}$ they were initialized to $[\alpha \cdot 10^{-3}, \alpha \cdot 10^{-4}, 10^{-8}]$ respectively.

Figure 4 shows our results. It is indeed the case across these different benchmarks (each of which has a different dataset, architecture, and optimizer type) that the taller the hyperoptimizer stack, the less sensitive the results become to the human-chosen hyperparameters. **With a three-level optimizer stack, a single hyperoptimizer design obtains reasonable results in all of our benchmarks across several orders of magnitude of base-level step size.**

**Further tests of scalability**    To test if our hyperoptimizers continue to work in even larger regimes, we fine-tuned a ResNet-152 (pretrained on ImageNet) to the Caltech-256 dataset Griffin et al. (2007). Figure 4e shows the results: a height-1 hyperoptimizer recovers $\approx 11\%$ error for both $\alpha = 10^{-6}$ and $\alpha = 10^{-4}$ (without a hyperoptimizer, $\alpha = 10^{-6}$ gives $91.5\%$ error). A height-2 hyperoptimizer is additionally able to make significant progress when $\alpha = 10^{-2}$.

We stress how lightweight and practical this method is. Figure 4f shows how runtime scales as a function of hyperoptimizer stack height for the above benchmarks. We find that the scaling is linear:

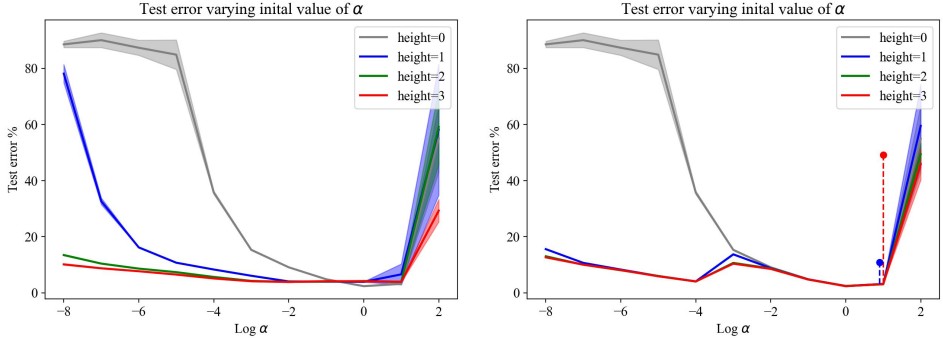

(a) Results on an MLP (Sec 3.1), where all layers are initialized with the same step size.

(b) Results on an MLP (Sec 3.1), where all layers are initialized as in Sec 3.4.

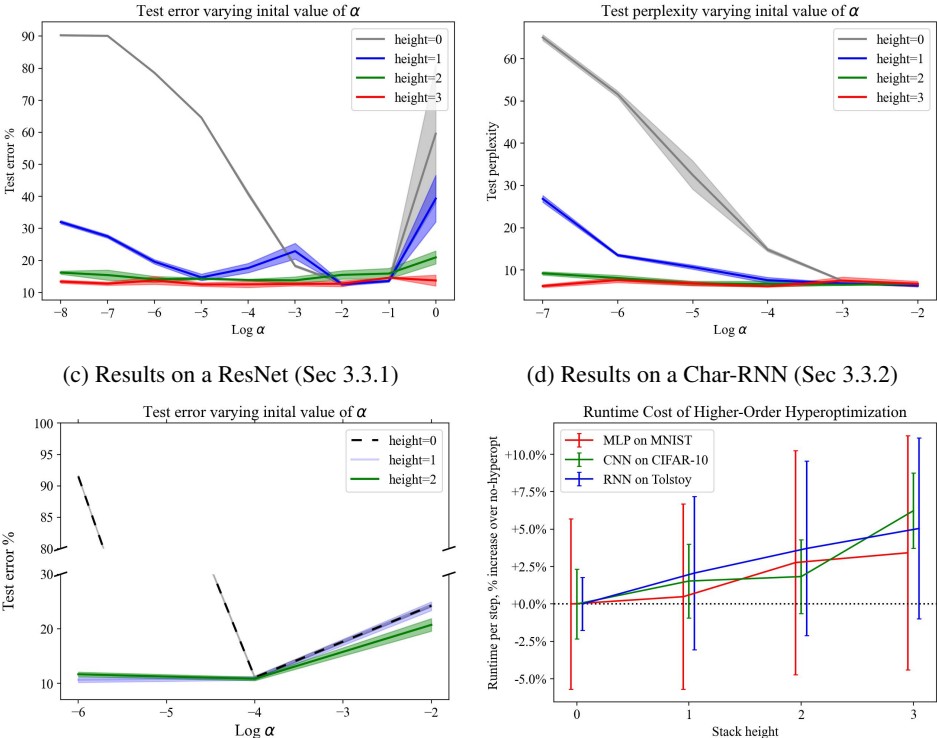

(c) Results on a ResNet (Sec 3.3.1)

(d) Results on a Char-RNN (Sec 3.3.2)

(e) Results on fine-tuning a pretrained ResNet-152 to the Caltech-256 dataset (Sec 3.4)

(f) Our hyperoptimizers have minimal impact on runtime, which scales linearly in height (Sec 3.4)

Figure 4: Evaluating higher-order hyperoptimization across a variety of benchmarks (Section 3.4). As we stack more layers of optimizers, the resulting hyperoptimizer is less sensitive to the initial choice of hyperparameters, but costs only 1-2% more in runtime.

**each additional level costs only 1-2% in additional runtime above the non-hyperoptimized baseline, and negligible additional memory.**

# 4   Related work

Hyperparameter optimization has a long history, and we refer readers to a recent survey by Feurer and Hutter (2019) for the full story. Most existing work on gradient-based hyperparameter optimization (Bengio, 2000; Domke, 2012; Maclaurin et al., 2015; Pedregosa, 2016; Franceschi et al., 2017) has focused on computing hyperparameter gradients after several iterations of training, which is computationally expensive. Baydin et al. (2018), building on a technique first published by Almeida

et al. (1999), propose instead updating hyperparameters at *each* step, and Rubio (2017) provides a convergence analysis. Wu et al. (2018) demonstrate a "short-horizon bias" in the related online "stochastic meta-descent" algorithm (Schraudolph, 1999), but Lichtarge et al. (2022) empirically find encouraging results when optimizing large sequence-to-sequence models. Luketina et al. (2016) apply a similar technique to regularization hyperparameters, though they note that their proposed method could work in principle for any continuous hyperparameter. Grefenstette et al. (2019) provide a library for metalearning via hypergradients. As discussed above, we expand upon this line of work in three directions: (1) by fully automating this process, rather than requiring manual derivative computations; (2) by optimizing hyperparameters beyond just the learning rate; and (3) by realizing the vision of recursive higher-order hyperoptimizers and evaluating the resulting algorithms. We find that they are indeed more robust to the initial human-chosen hyperparameter, which relates our work to other learning algorithms that minimize sensitivity to learning rates (Orabona and Tommasi, 2017; Vaswani et al., 2019).

## 5 Limitations and future work

As discussed in Section 3.4, one limitation of hyperoptimizers is that they cannot yet handle initial hyperparameters that are set far too high, because the system is unstable and diverges before the hyperoptimizer can have an effect. Designing hyperoptimizers robust in this regime requires further research, such as a deeper theoretical analysis of convergence. Our implementation also requires some care in avoiding certain bugs related to computation graph management. For example, loggers must detach what is logged to avoid memory leaks because tensors are not garbage collected unless all children are detached. Similarly, certain PyTorch modules (e.g. the built-in LSTM) cannot be used because they silently modify the computation graph, which may lead to incorrect gradients with our system. Further research is needed to design differentiable programming languages where methods like ours can be expressed in a modular and composable manner that minimizes the risk of such bugs.

**Broader impact**    Training a modern deep learning system consumes a tremendous amount of energy, and hyperparameter searches can multiply that energy impact by many orders of magnitude (Strubell et al., 2019). We hope that advances in on-line hyperparameter tuning can reduce this impact.

## 6 Conclusion

We presented a technique that enables gradient descent optimizers like SGD and Adam to tune their own hyperparameters. Unlike prior work, our proposed hyperoptimizers require no manual differentiation, learn hyperparameters beyond just learning rates, and can be stacked recursively to many levels. We described an elegant recursive implementation of hyperoptimizers in a reverse-mode AD system and evaluated it on a variety of benchmarks, showing that as the stacks grow taller, they become less sensitive to the initial human-chosen hyperparameter.

## Acknowledgments and Disclosure of Funding

We thank Samantha Andow, Emilio Arroyo-Fang, Irene Dea, Johann George, Melissa Grueter, Basil Hosmer, Steffi Stumpos, Alanna Tempest, and Shannon Yang for early discussions, Krishna Murthy Jatavallabhula and Josh Tenenbaum for their advice when preparing this paper, and the anonymous reviewers for their thoughtful feedback. KC and JRK were supported by NSF Grants #2105806, #CCF-1231216, #CCF-1723445 and #CCF-1846502, and ONR Grant #00010803 at MIT. Additionally, KC was supported by a Hertz Foundation Fellowship, the Paul and Daisy Soros Fellowship for New Americans, and an NSF Graduate Research Fellowship under Grant #2141064, and AX was supported by the MIT Undergraduate Research Opportunities Program (UROP).

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
