# OpenReview forum: "Gradient Descent: The Ultimate Optimizer"
_NeurIPS.cc/2022/Conference — NeurIPS 2022 Accept_

### Official Review · Reviewer_oELp · 2022-06-15

**Rating:** 7
**Confidence:** 4
**Soundness:** 3 good
**Presentation:** 2 fair
**Contribution:** 3 good

**Summary:**

This work presents a way of optimizing parameters (hyperparameters) of first order optimizers as the learning rate and the momentum coefficient using another first order optimizer as SGD or ADAM, called hyperoptimizer. Furthermore, this method can be applied recursively to optimize the parameters of the hyperoptimizer and so on, creating a chain of optimizers. The gradient of the loss w.r.t. the hyperparameters is computed by backpropagating through only one step of the optimizer, so that the weights and biases of the neural network and the hyperparameters can be optimized jointly with a small increase in cost compared to standard training.

The authors provide a simple pseudocode implementation in pythorch which exploits Automatic Differentiation (AD), thus avoiding error-prone implementation of derivatives by hand. They also validate this method on several neural architecture and datasets: MLPs on MNIST, CNNs on CIFAR-10 and RNNs on Tolstoy. Results show that this approach often outperforms training with the fixed sub-optimal hyperparameters used as a starting point for the algorithm, and is comparable to using the hyperparameters found at the end of training. Furthermore, they show that, when optimizing only the learning rate,  the longer the chain of optimizers the less sensitive the final performance is to the initial learning rate.


**Questions:**

1. How much higher is the time and memory cost of the method compared to standard SGD/ADAM? how does it increase as the chain of (hyper-)optimizers becomes longer?
The computational cost is discussed only briefly in the text and all plots have the number of epochs in the x-axis. I think this aspect is worth an additional table/section or the inclusion of a plot with time in the x-axis.

2. The authors should consider to specify that the function  $f$ in $f(w_i)$ and $f(w_{i-1})$ differ, since they are computed on different minibatches. This could be done by e.g. using $f_i$ in place of $f$. I think this would make the exposition clearer.

3. Did the authors try to theoretically study the convergence of the proposed approach?
I think this could be an interesting avenue for future work and could be discussed in the paper as a limitation of the current work.

4. How many runs have been done for each experiment? How are errors/variance in the table/plots computed? This should be discussed when presenting the result or in the appendix.


**Limitations:**

The authors properly discussed the limitations of the method in Sec. 5. Main limitations are handling high learning rate values and implementation issues related to the PyTorch implementation.

**Strengths And Weaknesses:**

Strenghts:
1. The method is simple to implement, clearly explained and probably not too costly compared to standard optimizers. The authors present Pytorch code snippets and discuss the approach in great detail.
2. Convincing experimental evaluation. Experiments are done with several architectures, MLPs CNNs and RNNs, and small to medium sized dataset. The advantages brought by the method are evident and match the authors’ claims.
3. High significance. A Better optimizer for training neural networks is useful to the majority of the machine learning community. The ease of implementation could favor its adoption.

Weaknesses:
1. Comparison with related works can be improved. The authors discuss a connection with programming language theory without providing references. They also do not discuss or compare against other adaptive optimizers, as [1, 2], which are also less sensitive to the initial learning rate and are more theoretically grounded.
2. Lacks large scale experiments (e.g. on ImageNet) and does not include transformers models, which are widely used neural architectures.
3. Limited Novelty. The idea of optimizing the learning rate using a first order method with gradient computed by back propagating through one step of the optimizer was presented in [3]. The novelty of this work lies in (i) applying the idea recursively, (ii) optimizing other hyperparameters like the momentum coefficient, (iii) providing a simple Pytorch implementation exploiting automatic differentiation and (iv) an independent experimental evaluation.

[1] Orabona, Francesco, and Tatiana Tommasi. "Training deep networks without learning rates through coin betting." Advances in Neural Information Processing Systems 30 (2017).

[2] Vaswani, Sharan, et al. "Painless stochastic gradient: Interpolation, line-search, and convergence rates." Advances in neural information processing systems 32 (2019).

[3] Baydin, Atilim Gunes, et al. "Online Learning Rate Adaptation with Hypergradient Descent." International Conference on Learning Representations. 2018.

**Post author's response**
The authors addressed most of my concerns and promised to address others. Therefore I am increasing my score from 6 to 7.

---

> ### Author Response · Authors · 2022-08-02
> **Authors' response to review from oELp**
>
> Thank you for your thoughtful review of our work. We discuss novelty, large-scale experiments, and computational cost in our common response above. Below, we address the other key points you raised.
>
> **Related work:** We will cite and discuss the relevant programming language theory we take inspiration from [e.g. 1-5] when revising. This line of work broadly argues that careful self-application of simple ideas (e.g. functional abstraction or partial evaluation) unlocks unexpected power. Here, we apply the same philosophy to gradient descent optimization via automatic differentiation.
>
> [1] Steele, G and Sussman, G. _Lambda: The Ultimate Imperative._ 1976.
> [2] Pearlmutter, B and Siskind, J. _Lambda: The Ultimate Backpropagator._ 2008.
> [3] Wang, F et al. _Shift/reset: The Penultimate Backpropagator_. 2019.
> [4] Futamura, Y. _Partial computation of programs_. 1983.
> [5] Curry, H. _Combinatory logic._ 1958.
>
> We will also discuss our work's relationship with related adaptive optimizers as suggested. COCOB and SGD-Armijo indeed also seek to reduce effort spent tuning step sizes. Our method is different because it is a simple, lightweight enhancement on top of _existing_ widely-used optimizers (e.g. Adam). This makes it an appealing option for practitioners already invested in standard PyTorch-style training infrastructure.
>
> **Subscripting $f_i$:** This suggestion is well-taken; we will adjust the notation accordingly when revising.
>
> **Theoretical study of convergence:** While we do not embark on a theoretical treatment of convergence here, recent work [6] has explored this question with promising early results, including a proof of convergence of Baydin et al's single-level hyperoptimizer for quadratic functions. We will cite and discuss these results when revising.
>
> [6] Rubio, D. _Convergence Analysis of an Adaptive Method of Gradient Descent._ 2017. https://damaru2.github.io/convergence_analysis_hypergradient_descent/dissertation_hypergradients.pdf
>
> **Number of runs for each experiment:** We report the mean and standard deviation for 3 runs of each experiment throughout the paper. We will update the text to mention this.

---

> > ### Comment · Reviewer_oELp · 2022-08-08
> > **Thanks for the additional experiments and the detailed response!**
> >
> > The authors addressed some of my concerns and promised to address others. I am also impressed by the amount of effort in the author's response. I updated my review and increase the score accordingly.

---

### Official Review · Reviewer_zqnY · 2022-07-06

**Rating:** 7
**Confidence:** 5
**Soundness:** 4 excellent
**Presentation:** 4 excellent
**Contribution:** 4 excellent

**Summary:**

This paper outlines a method to computer hypergradients using automatic differentiation by modifying backpropagation, allowing hypergradient descent to be extended to arbitrarily large layers of hypergradient descent without consuming arbitrarily large amounts of programming labor. The paper then uses this technique to validate the hypothesis that taller towers of hyperoptimization yield increasingly robust hyperparameters, eventually converging to the optimal as the number of layers increases even with poorly chosen initial values.

**Questions:**

- The computational overhead of hypergradient descent using AD is stated to be negligible; what is the actual value –  is it even statistically measurable?


**Limitations:**

Authors address all limitations I can think of.


**Strengths And Weaknesses:**

Strengths:
- Paper is extremely well written, and the best that I have reviewed recently. The writing is very easy to understand and follow.
- I believe that the experimental results validating the higher order hyperoptimization hypothesis will be immensely valuable to the optimization and meta-learning community.
- Well-documented, easily readable, and easily usable source code is provided.
- The method presented will be very useful to the community in advancing future meta-learning and optimization research.

Weaknesses:
- The contributions of this paper are primarily in presentation, and not as deep or novel as the typical NeurIPS paper.
- Evaluation networks are rather small compared to networks used in modern practice, and more comparable to those used by learning to optimize papers that are crippled by high memory and compute requirements. Since the overhead of hypergradient descent is much lower, it would be good to see a benchmark of at least a ResNet-152 or some transformer architecture, especially in a higher-order hyperoptimization experiment.

---

> ### Author Response · Authors · 2022-08-02
> **Authors' response to review from zqnY**
>
> Thank you for your thoughtful review of our work. For our remarks on novelty, large-scale experiments, and computational cost, please see our common response above. We look forward to discussing further during the next phase of the review process.

---

### Official Review · Reviewer_kxHt · 2022-07-11

**Rating:** 6
**Confidence:** 4
**Soundness:** 3 good
**Presentation:** 4 excellent
**Contribution:** 3 good

**Summary:**

This paper proposes to apply reverse mode auto-differentiation to compute gradients w.r.t  hyperparameters of an optimizer and update them using gradient descent, in contrast to computing them manually as done in previous works. It offers the flexibility in terms of choosing the optimizer for updating the hyperparameters. It also proposes to apply this procedure recursively for each hyperparameter of hyperparameters, resulting in a stacked hyperoptimizer. Experimentally, it tests the hyperoptimization idea on the hyperparameters of SGD and Adam, and compare its performance against the case when hyperoptimization is not used.


**Questions:**

- To strengthen the paper, it would be good if the author can further test the stacked hyperoptimizer on more optimizers, such as AdaGrad, RMSProp etc, and report similar results as Figure 2 and 4.
- It would be good to plot also the training loss of stacked and unstacked hyperoptimizers. It is interesting to see whether the hyperoptimizer converges faster than normal SGD or Adam. Also please report the computation time for different heights of the stacked hyperoptimizer.
- I believe the experiments of this work are interesting and intriguing, and can be beneficial for the community to reduce the search cost of hyperparameter or build better optimizers. This paper is at the border line of acceptance. If all my concerns are addressed, I would raise my score.

**Ethics Review Area:**

["I don’t know"]

**Strengths And Weaknesses:**

- This paper is well-written and can be easily understood. The pseudocode together with the computation graph clearly demonstrate the hyperoptimization procedure. There is a minor typo in section 3.1, should be “4.08% and 4.48% test error for SGD/SGD and SGD/Adam, respectively”. The main findings are also highlighted.
- The novelty of this paper is limited. As mentioned in the paper, [1] computed the derivative of loss w.r.t. the learning rate of SGD and performed gradient-descent type update. For optimizers with more hyperparameters and more complicated gradients such as Adam, the paper does not demonstrate the clear advantage of optimizing all the hyperparameters over only optimizing the learning rate. For example, in Table 1 (b), when only the learning rate of Adam is optimized, the resulting test error is lower than that when all the hyperparameters are optimized (3.00 % vs 3.03 %). Hence, autodifferentiation may not take full advantage over manual computation when simple expression for updating the learning rate can be derived as in the case of SGD or Adam.
- Despite lack of novelty, some interesting results are presented. Figure 1a) shows that when hyperoptimization is used, the algorithm is less sensitive to the initialization of hyperparameters, and can make certain degree of corrections when bad initialization occurs. This is an intriguing property as a good initialization is difficult to know a priori and can be dependent on many factors such as optimizer and dataset. Another interesting observation is that the final step size can be either larger or smaller when compared against its initial value as shown in Table 1 (a) and Figure 2 (b). This shows that the hyperoptimization approach can tune the hyperparameter properly depending on its magnitude.
- One main contribution of this paper is that with autodifferentiation, one can perform gradient -type optimization for the hyperparameter of hyperparameters. The results seem to show that test error decreases as height increases, but it is not very convincing when comparing Figure 4 b) with Figure 4 c). The claim regarding the sensitivity of different height to base-level step size does seem to be true.
- Some main limitations of the experimental results are: given hyperparameters are properly initialized, the hyperoptimization does not demonstrate clear advantage over the case when they are fixed during training. Another limitation is that even hyperoptimizer can optimize hyperparameters of hyperparameters, but one still need to provide proper initializations for these higher-level hyperparameters. Hence, the search cost of hyperparameter may not be necessarily reduced. Moreover, as mentioned in the paper, it is difficult to provide a common initialization for different heights. This may limit the usage of the stacked hyperoptimizer.

[1] Online Learning Rate Adaptation With Hypergradient Descent

---

> ### Author Response · Authors · 2022-08-02
> **Authors' response to review from kxHt**
>
> Thank you for your thoughtful review of our work. For our remarks on novelty, large-scale experiments, and computational cost, please see our common response above. Below, we address the other key points you raised.
>
> ---
>
> ### AdaGrad and RMSProp
>
> Because of our core contribution, using automatic differentiation instead of manual differentiation to compute hypergradients, we were easily able to implement and test AdaGrad and RMSProp in a matter of minutes. We replicated the experiments of Table 1 with these two optimizers, validating that both AdaGrad and RMSProp benefit from hyperoptimization. This is the case both with SGD and the same optimizer _itself_ on top. (Here, "…" indicates default hyperparameters from PyTorch documentation, $\alpha=0.01, \epsilon=10^{-10}$ for AdaGrad and $\alpha=0.01, \epsilon=10^{-8}, \gamma=0.99$ for RMSProp.)
>
> | AdaGrad experiments | Test Error |
> |--------------------------------------|------------|
> | SGD(0.01)                            | 8.99±0.22% |
> | SGD(0.01)/AdaGrad(…)          | 4.40±0.07% |
> | AdaGrad(…)                    | 3.71±0.04% |
> | AdaGrad(…)/SGD(1e-3)          | **3.16±0.08%** |
> | AdaGrad(…)/AdaGrad(…)  | 3.58±0.14% |
>
> **Remark:** AdaGrad is known to push its step size to zero over time, which eventually stalls learning. Looking at the hyperoptimizer's learned step size over time, we see that our hyperoptimizer indeed gradually raises the step size over time to make up for this effect.
>
> | RMSProp experiments | Test Error |
> |--------------------------------------|------------|
> | SGD(0.01)                            | 8.99±0.22% |
> | SGD(0.01)/RMSprop(...)               | 4.00±0.14% |
> | RMSprop(...)                         | 3.03±0.16% |
> | RMSprop(...)/SGD(1e-3)               | 2.88±0.08% |
> | RMSprop(...)/RMSprop(...)            | 2.98±0.17% |
> | RMSprop(...)/SGD{also learning $\gamma$}(1e-3, 1e-5) | **2.78±0.03%** |
>
> **Remark:** RMSprop has a new hyperparameter $\gamma$, which modulates the accumulation of gradient RMS terms. Our system makes it trivial to automatically learn $\gamma$ alongside the learning rate, which yields positive results as shown in the last row of the table.
>
> ---
>
> ### Limitations of experimental results
>
> * _"Hyperoptimization does not demonstrate clear advantage over fixed, optimally-initialized hyperparameters":_ This is technically true; however, the goal of the hyperoptimizer is not to improve upon _known_ optimal settings but rather to help discover these optimal settings when they are _unknown_. Indeed, for sub-optimal initializations hyperoptimization does demonstrate clear advantage. Furthermore, even with optimal initialization, our method can improve upon fixed hyperparameters by automatically finding a good decay schedule (Figure 2a).
> * _"Difficult to provide a common initialization for different heights":_ We emphasize that _all_ of the results reported in Figure 4(b,c,d) were produced with the same initialization settings.
> * We stress the value to the field of _any_ experimental evaluation of higher-order hyperoptimizers, which were only theorized by prior work (Baydin et al, 2018). Our core contribution of reducing general gradient-based hyperoptimization to automatic differentiation was key to making this first experimental evaluation of higher-order hyperoptimization possible.
>
> ### Training loss
>
> We will include training loss curves for all experiments in our revision. In short, they consistently match the story told by the test error curves (hyperoptimizer has comparable convergance for "good" hyperparameter initializations and significantly better convergance for "bad" hyperparameter initializations).

---

> > ### Comment · Reviewer_kxHt · 2022-08-03
> > **Update**
> >
> > Thank you for the detailed response and effort.

---

### Author Response · Authors · 2022-08-02
**Common response to all reviewers**

Thank you all for your thoughtful reviews. Here, we respond to some common concerns across all reviewers.

---

### Computational cost

This table presents runtimes per epoch for our three benchmarks, organized by height of hyperoptimizer stack (each trial is run on a single NVIDIA TITAN X GPU). **The percent increase over the no-hyperopt baseline is just 1-2% for each additional layer of hyperoptimization.** We will include this information in a revision, along with training curves showing time on the X-axis as suggested by reviewer oELp.

| Experiment            |            No hyperopt |               Height=1 |              Height=2 |               Height=3 |
| :-------------------- | ---------------------: | ---------------------: | --------------------: | ---------------------: |
| MLP on MNIST          |   6.14±0.35 s (+0.00%) |   6.17±0.38 s (+0.49%) |  6.31±0.46 s (+2.77%) |   6.35±0.48 s (+3.42%) |
| Resnet-20 on CIFAR-10 |  20.21±0.47 s (+0.00%) |  20.52±0.50 s (+1.48%) | 20.58±0.50 s (+1.83%) |  21.47±0.51 s (+6.23%) |
| Char-RNN on Tolstoy   | 80.25±1.42 ms (+0.00%) | 81.91±4.11 ms (+2.25%) | 83.23±4.68 ms (+4.0%) | 84.30±4.85 ms (+5.38%) |

---

### Novelty and contributions

The idea of online gradient-based learning rate adaptation originated with Almeida et al (1998) and was applied to modern deep neural networks 20 years later by Baydin et al (2018). Our contributions in this paper are key to advancing this line of work, by showing how to correctly use automatic differentation to enable efficient and generic self-application of optimizers. (Recall that a naive AD implementation, without proper computation graph management, suffers quadratic-time slowdown.)

Our contribution dramatically reduces the manual effort needed to generalize the algorithm to other optimizers and to other hyperparameters beyond learning rate. For example, in response to Reviewer kxHt's question, we added experiments on AdaGrad and RMSProp within minutes, even optimizing RMSProp's second hyperparameter $\gamma$. We are also able to provide the first empirical study of higher-order hyperoptimizers, which Baydin et al could only theorize. Finally, our "plug-and-play" implementation in PyTorch, which we will make publicly available, enables immediate practical adoption by the community.

Almeida L et al. _Parameter adaptation in stochastic optimization._ On-Line Learning in Neural Networks. 1998.

---

### Large-scale experiments

We chose our experiments to match those of Baydin (2018) in scope, but are happy to run large-scale experiments (e.g. ResNet-152 on ImageNet, or a transformer) for a final revision. While we do not have enough time to train very large models from scratch during the 1-week author response period, we can show preliminarily that our method scales by demonstrating its performance on a fine-tuning task.

Here, we fine-tune ResNet-152 pretrained on ImageNet to the Caltech-256 dataset. We vary the initial learning rate $\alpha$ and show that hyperoptimizers indeed recover good test error across these conditions, reducing the sensitivity of the system to the initial choice of $\alpha$ as desired. A height-1 hyperoptimizer recovers ≈11% test error for both $\alpha=10^{-6}$ and $\alpha=10^{-4}$. A height-2 hyperoptimizer is additionally able to make significant progress when $\alpha=10^{-2}$.

| Initial Hyperparameters   | Test error - No Hyperopt | Height=1    | Height=2    |
| ------------------------- | ------------------------ | ----------- | ----------- |
| $\alpha=10^{-6}, \mu=0.9$ | 91.51±0.21%              | 10.66±0.59% | 11.67±0.50% |
| $\alpha=10^{-4}, \mu=0.9$ | 11.08±0.35%              | 10.94±0.43% | 10.84±0.41% |
| $\alpha=10^{-2}, \mu=0.9$ | 24.24±0.32%              | 24.15±0.98% | 20.73±1.41% |

---

### Public Comment · ~Core_Francisco_Park1 · 2023-05-25
**MNIST Adam Basline Accuracy seems lower than what I get**

Hi,

I was reading this work and trying to use it for my research purposes. First of all, thanks for the great work.

Unfortunately, I am not seeing a improvement in my test accuracy using this optimizer on MNIST. I decided to experiment on the code you have on github (since my own model uses ReLU).

With the pytorch's default Adam optimizer, I get the following result (cropped):

EPOCH: 19, TRAIN LOSS: 0.8249346494356791EPOCH: 19, TEST ACC: 0.9741
EPOCH: 20, TRAIN LOSS: 0.823439075311025EPOCH: 20, TEST ACC: 0.9743
EPOCH: 21, TRAIN LOSS: 0.821824014822642EPOCH: 21, TEST ACC: 0.9736
EPOCH: 22, TRAIN LOSS: 0.8204631138801575EPOCH: 22, TEST ACC: 0.9745

I am wondering why my test error is smaller than any of the test errors in Table 1, especially the baseline Adam (which I am directly comparing to).

Could you provide code reproducing the results in Table 1? Thanks!

I am happy to provide code if needed!

I apologize if this is not the right way to ask for feedback about the paper, I am new to OpenReview.
Thanks!!

---

### Meta-Review · Area_Chair_GQ7u · 2022-08-25

**Recommendation:** Accept
**Confidence:** Certain

**Metareview:**

All reviewers recommend accepting the paper. Many years ago I experimented with this type of approach, for a single layer of hyper-parameters, and came to the conclusion that setting the hyper-hyper-parameters was just as difficult as setting the parameters of the optimization algorithm. I am shocked that this changes as you increase the number of layers, and I think others would be too. If this approach genuinely reduces hyper-parameter tuning in such a simple way, it should be a spotlight or oral presentation.

**Award:**

Yes

---

### Decision · Program_Chairs · 2022-09-14

Accept